# Mechanisms of Atrial Fibrillation in Obstructive Sleep Apnoea

**DOI:** 10.3390/cells12121661

**Published:** 2023-06-19

**Authors:** James Saleeb-Mousa, Demitris Nathanael, Andrew M. Coney, Manish Kalla, Keith L. Brain, Andrew P. Holmes

**Affiliations:** 1School of Biomedical Sciences, Institute of Clinical Sciences, University of Birmingham, Edgbaston, Birmingham B15 2TT, UK; dxn670@student.bham.ac.uk (D.N.); a.m.coney@bham.ac.uk (A.M.C.); k.l.brain@bham.ac.uk (K.L.B.); 2School of Biomedical Sciences, Institute of Cardiovascular Sciences, University of Birmingham, Edgbaston, Birmingham B15 2TT, UK; manish.kalla@uhb.nhs.uk; 3Queen Elizabeth Hospital, Birmingham B15 2GW, UK

**Keywords:** atrial fibrillation, obstructive sleep apnoea, chronic intermittent hypoxia, autonomic nervous system

## Abstract

Obstructive sleep apnoea (OSA) is a strong independent risk factor for atrial fibrillation (AF). Emerging clinical data cite adverse effects of OSA on AF induction, maintenance, disease severity, and responsiveness to treatment. Prevention using continuous positive airway pressure (CPAP) is effective in some groups but is limited by its poor compliance. Thus, an improved understanding of the underlying arrhythmogenic mechanisms will facilitate the development of novel therapies and/or better selection of those currently available to complement CPAP in alleviating the burden of AF in OSA. Arrhythmogenesis in OSA is a multifactorial process characterised by a combination of acute atrial stimulation on a background of chronic electrical, structural, and autonomic remodelling. Chronic intermittent hypoxia (CIH), a key feature of OSA, is associated with long-term adaptive changes in myocyte ion channel currents, sensitising the atria to episodic bursts of autonomic reflex activity. CIH is also a potent driver of inflammatory and hypoxic stress, leading to fibrosis, connexin downregulation, and conduction slowing. Atrial stretch is brought about by negative thoracic pressure (NTP) swings during apnoea, promoting further chronic structural remodelling, as well as acutely dysregulating calcium handling and electrical function. Here, we provide an up-to-date review of these topical mechanistic insights and their roles in arrhythmia.

## 1. Clinical Associations between AF and OSA, and Current Treatment Strategies

### 1.1. OSA Predisposes to AF

AF is the most common sustained cardiac arrhythmia with an approximate global prevalence between 2 and 4%, and it is a significant healthcare burden [1,2]. AF is associated with an increased risk of stroke, heart failure, cognitive decline, depression, hospitalisation, and death [2]. 

OSA is a sleep-related breathing disorder characterised by intermittent upper airway collapse leading to recurrent apnoeas and hypopnoeas. Apnoeic events are classified as a greater than 90% reduction in baseline airflow for a period of 10 s or more, whilst hypopnoeas are defined as at least a 50% decrease in baseline airflow for a period of 10 s or more, accompanied by a 3–4% O_2_ desaturation or microarousal [3]. OSA severity is stratified on the basis of the number of apnoeic/hypopnoeic events that take place per hour (apnoea hypopnoea index, AHI); with ≥5 to <15 classed as mild, ≥15 to <30 as moderate, and ≥30 as severe [3,4]. Key features of OSA include episodic exposure to hypoxia and hypercapnia, numerous microarousals, repetitive autonomic nervous system activation and large swings in intrathoracic pressure as the patient makes exaggerated respiratory efforts against an occluded upper airway [3].

Importantly, OSA is comorbid in 21 to 74% of AF patients and is associated with an 88% increased incidence of AF [5,6]. This relationship is supported in part by common risk factors, such as age, hypertension, and cardiovascular disease; therefore, the independent attributable risk has been reported as 21% (OR 1.12–1.31) [7,8]. A dose-dependent response is seen with AF risk and increasing OSA severity [9]. Additionally, OSA status confers a greater risk of hospitalisation and worsening AF-related symptom severity, and it increases the risk of cardiovascular events [10,11]. This may also be partially attributable to the fact that OSA reduces the effectiveness of AF management. For example, OSA status increases the likelihood of AF recurrence following ablation and cardioversion [12,13,14,15]. Overall, a large body of clinical evidence supports the involvement of OSA in the pathogenesis and maintenance of AF.

### 1.2. Clinical Management of AF in Patients with OSA

The 2020 ESC guidelines recommend screening for OSA in patients with AF, reflecting consensus opinion that optimal OSA management may play a role in reducing the AF burden [2]. CPAP is an established means of reducing OSA severity and has been shown to protect against AF incidence, recurrence, and progression as well as alleviate antiarrhythmic drug (AAD) dependence in OSA patients [10,12,14,16,17,18,19,20]. Indeed, poor CPAP compliance and under-diagnosis of OSA means that these associations are likely to be conservative [17].

Patients with AF and OSA exhibit an increased risk of recurrence post ablation and an increased prevalence of non-pulmonary vein triggers [13,21,22]. The relationship between OSA and the AF substrate characterised by low left atrial voltage was recently studied by Nalliah et al. using high-density mapping [23]. They observed a dose-dependent association of OSA severity with the level of conduction heterogeneity, voltage heterogeneity, and the number of low voltage areas, which was consistent across all regions of the left atrium. In this study, low voltage was determined as a bipolar voltage of less than 0.5 mV. The AHI was also associated with lower overall left atrial voltage but not with slower conduction velocity. Patients with paroxysmal AF and OSA were suggested to have discrete low-voltage zones combined with a slow conduction velocity as the atrial substrate. In patients with persistent AF, there appeared to be a more diffuse pattern of low voltage across the entire left atrium. The same group was recently able to test the impact of CPAP on this substrate in a randomised controlled trial [24]. CPAP therapy reversed atrial remodelling with progressive changes in the untreated arm; however, a meta-analysis did not extend this observation to improved results in patients undergoing ablation [25]. The protective effects of CPAP appeared to be greater in younger, obese patients, in whom the extent of cardiac remodelling is likely to be less advanced [19].

It was recently shown that effective early rhythm control therapy with either ablation or AADs decreased the risk of adverse cardiovascular outcomes in patients with newly diagnosed AF [26]. Therefore, an emerging challenge is to achieve optimal and more personalised rhythm control therapy to ensure long-term sinus rhythm. Approximately 10–20% of patients with AF are treated with AADs [2,27]. These medications are highly effective in some patients but do not work in others, and AF recurrence remains unacceptably high [2]. The variable efficacy likely reflects the numerous underlying disease mechanisms that can lead to AF development and its progression [28]. An important research need is to better characterise the specific disease mechanisms underpinning AF development associated with genetic modifications and/or co-morbidities, including OSA. 

Preclinical data have suggested that the effectiveness of flecainide and dronedarone can be predicted on the basis of the expression level of key AF-associated genes, such as *PITX2*, or the status of the atrial resting membrane potential, which is often more negative in AF patients [29,30]. At present, specific information regarding the efficacy of AADs in patients with AF and OSA is scarce. In a small cohort of 61 patients, it was observed that those with severe OSA were less responsive to AAD treatment compared with those without OSA and those with more mild/moderate OSA [31]. In a follow-up pilot study, it was reported that the presence of AF-associated risk variants adjacent to the *PITX2* gene reduced the effectiveness of AADs by the same amount as severe OSA [32]. Interestingly, the presence of these risk variants combined with severe OSA did not further deplete the sensitivity of the AADs, possibly indicating a shared pathway between the two. Large-scale clinical trials are warranted to evaluate the effectiveness of individual AADs in OSA patients and better understand the interaction between OSA and AF-associated genetic variants. Furthermore, there is a potential to develop new AADs specifically for patients with OSA and AF that target the underlying disease mechanism(s). This strategy should lead to better selection of AADs and more effective and personalised rhythm control therapy in patients with OSA and AF. 

## 2. Arrhythmogenic Mechanisms in OSA

The need for integrative AF management and more effective AAD treatment has driven an expanding body of research aiming to characterise the mechanisms of atrial arrhythmogenesis in OSA. Broadly, these studies have aimed to address the cardiac remodelling processes concerned with the structural, electrical, and autonomic properties of the atria. The pathological mechanisms are based on three key features of OSA: (1) autonomic imbalance, (2) NTP swings, and (3) CIH.

### 2.1. Autonomic Imbalance

Cardiac electrical function is subject to modification by the interplay between the sympathetic and parasympathetic arms of the autonomic nervous system. In OSA patients, there are rapid transitions in cardiac autonomic balance which are dependent on the different phases of the acute apnoeic episodes. In the apnoeic period itself, there is substantial bradycardia, which is indicative of parasympathetic dominance [33]. Immediately after the apnoea, there is a powerful rebound tachycardia, which is likely due to parasympathetic withdrawal uncovering an increase in cardiac sympathetic activity. The heart rate then gradually returns to the resting level [33]. Similarly, arterial blood pressure also rises immediately after the apnoea, suggesting an acute rise in vascular sympathetic activity [34]. The precise mechanisms underpinning these changes in autonomic activity are poorly understood but are likely to have contributions from arterial baroreceptors, peripheral chemoreceptors, central respiratory centres, and pulmonary afferents. Furthermore, it is still not clear whether atrial arrhythmias in humans are directly initiated by these acute changes in cardiac autonomic activity and whether there is an increased risk either during or immediately after the apnoea.

In addition to acute alterations in autonomic activity during the apnoeic episodes, there are also more chronic changes that emerge over time. Pulse/heart rate variability (PRV/HRV) are useful clinical measures of autonomic balance. Measures of HRV reveal a transition to sympathetic predominance in OSA patients during sleep, manifesting as an increased low frequency/high frequency (LF/HF) ratio [35]. Other studies have found that an increasing nocturnal PRV is an independent predictor of AF risk [36]. Chronic changes in cardiac autonomic imbalance in OSA are likely to be underpinned by both qualitative and quantitative changes in cardiac autonomic innervation. For instance, animal models of OSA are associated with an atrial hyperinnervation of both the sympathetic and parasympathetic fibres but with an overall preponderance towards the sympathetic system. This is represented both in terms of raw fibre density and the expression of adrenergic/cholinergic receptors in cardiomyocytes [37,38,39]. Cardiac autonomic growth in OSA is further supported by findings of increased growth-associated protein 43 at the neuronal growth cones and signs of neuronal sprouting, which may be a result of increased NGF expression [37,40]. In AF patients, a similar atrial sympathetic hyperinnervation has been shown [41,42], with Deneke et al. also noting downregulation in atrial cholinergic innervation. These studies demonstrate the role of autonomic imbalance in creating a vulnerable AF substrate. Interestingly, extrinsic autonomic denervation in OSA animal models suppresses these trophic changes, suggesting that they may be a result of chronic overstimulation. The resulting autonomic hyperinnervation has been demonstrated to increase AF inducibility and duration in CIH models [38,39].

Despite a predominant sympathetic hyperinnervation, AF initiation during apnoeic episodes seems to be more potently driven by acute rises in parasympathetic activity. For instance, the well-documented shortening of the atrial effective refractory period (AERP) observed in OSA models is significantly more attenuated by muscarinic blockade than by β-adrenoceptor blockade. This manifests as significantly greater decreases in AF inducibility and duration, as well as ectopic activity and recurrence [39,43]. Indeed, preferential sensitivity to parasympathetic activation in AF is also seen in the pacing of healthy animal models; however, the difference is significantly more pronounced in OSA [44]. It is unclear whether this may be a result of differential muscarinic/adrenergic receptor expression density; some groups have reported a predominance of β_1_ adrenoceptors, and others have reported a predominance of M_2_ receptors [39,44]. Recently, more consistent reports have been made on the variations in ion channel expression in OSA models. Of particular interest is the downregulation of the Ca_v_1.2 subunit of the L-type voltage-gated calcium channel responsible for the inward Ca^2+^ current, I_Ca,L_, and the upregulation of the G-protein-gated potassium channel responsible for the acetylcholine-activated inward rectifier potassium current, I_KACh_ [39,45,46,47]. I_KACh_ is positively modulated by M_2_ receptor activation and represents an important mechanism of parasympathetic vagal cardiac control by contributing to hyperpolarisation and action potential duration shortening, two key features of AF electrical remodelling [48]. Conversely, I_Ca,L_ is positively modulated by the β_1_-adrenoceptor and contributes to action potential duration by increasing the repolarisation delay as well as mediating excitation–contraction coupling [49]. Overall, it seems feasible that a chronic predominance in sympathetic tone secondary to hyperinnervation in OSA is met with a cellular adaptation in favour of counter-balancing parasympathetic signalling. This, in turn, could accentuate the transient increase in AF risk when stimulated by acute parasympathetic outflow during apnoea.

Indirectly, sympathetic activity may also contribute to atrial remodelling through the activation of the renin-angiotensin system. Aliskiren, a direct renin inhibitor, and eplerenone, a mineralocorticoid receptor antagonist, have separately been shown to attenuate OSA-induced reductions in I_Ca,L_ current density as well as alleviate AF inducibility [45,46]. It is unclear, however, whether the effect of these drugs on AF inducibility is a direct cellular effect or a secondary effect on cardiovascular function. Further characterisation of the signalling processes underpinning this remodelling may provide novel therapeutic targets for OSA-related AF. 

Transient Ca^2+^ ‘sparks’ are a key mechanism contributing to AF induction associated with sympathetic activation. These may precipitate ectopic activity by generating delayed after-depolarisations through activation of the sodium–calcium exchanger [50]. Interestingly, despite sympathetic hyperinnervation and acute apnoeic sympathetic activity being established hallmarks of OSA, a recent report found no significant increase in Ca^2+^ transient amplitude in a rat model of OSA, despite showing increases in expression of the Ca^2+^ handling proteins RyR2 and CAMKII [47]. However, this may reflect the absence of sympathetic stimulation in this study. Furthermore, studies using OSA models have not investigated the SR Ca^2+^ load, Ca^2+^ transient decay time, Ca^2+^ spark frequency, or the risk of delayed after-depolarisations, all of which could be important in initiating arrhythmia and be exaggerated in response to sympathetic stimulation. 

Autonomic hyperactivity in OSA has also been shown to be associated with atrial structural remodelling, creating a vulnerable substrate for re-entry. For instance, OSA-induced cardiomyocyte necrosis, oedema, and connexin (CX) 43 downregulation are preventable by cardiac sympathetic denervation in rats [38]. In similar studies, metoprolol was found to prevent atrial fibrosis and cardiomyocyte apoptosis, and similar findings in the ventricles can be explained by the inhibition of MAPK [42,51]. The role of MAPK/ERK signalling is well-established in apoptosis and fibrosis [52,53]. Further evidence points towards this signalling process being upregulated in OSA, as tolvaptan and doxycycline have recently been found to inhibit atrial fibrosis and matrix metalloproteinase (MMP) 9 expression in OSA rats by altering ERK signalling [54,55]. β-adrenoceptor-mediated activation of MAPK/ERK may therefore represent an important mechanism of structural remodelling mediated by autonomic hyperactivity. Interestingly, however, carotid body (CB) ablation, despite restoring blood pressure control and PRV in another rat model of OSA, did not prevent structural remodelling despite decreasing sympathetic cardiac outflow [56]. Furthermore, fibrosis and oxidative stress were unchanged in an OSA pig model following renal sympathetic denervation [57]. Therefore, whether these features of structural remodelling in OSA are mediated more predominantly through the local cellular effects of hypoxia rather than by autonomic signalling remains uncertain. Further investigation is thus required to elucidate these emerging cellular mechanisms of crosstalk between the autonomic system and the tissue microenvironment. It should also be considered that sympathetic overactivation may act to promote atrial structural remodelling simply through the metabolic stress associated with increased cardiac work.

### 2.2. NTP Swings

NTP swings of approximately −10 to −15 mmHg are generated during apnoea due to the movement of the respiratory muscles against an occluded airway [58]. In a rat model of OSA, respiratory muscle paralysis during apnoea was found to be protective against AF inducibility [59]. NTP generated by respiratory muscle action is sufficient to elicit a significant parasympathetic response, likely due to baroreceptor activation, as well as contribute to atrial stretch. Both of these mechanisms could act to initiate atrial arrhythmia during acute apnoea.

NTP produces a demonstrable vagal activation that is attenuated by anticholinergics and vagotomy and does not occur with CIH alone [59,60]. The effects of NTP in these models are comparable to high-level baroreceptor stimulation and significantly increase AF inducibility by reducing AERP, as illustrated in Figure 1 [61]. These data support the view that acute parasympathetic reflex activation precipitates AF during apnoea. Interestingly, low-level baroreceptor stimulation abolishes the effect of NTP on AERP and AF inducibility [61]. This is consistent with reports of the protective effects of low-level vagal stimulation in other OSA and non-OSA animal models [62,63,64], as well as in AF patients by stimulation of the tragus [65,66]. Low-level vagal stimulation may therefore represent an attractive therapeutic approach for OSA-related AF. 

NTP also exerts direct mechanical stress on the atria. Left atrial stretch predisposes to AF via both acute and chronic atrial remodelling processes. Acute stretch, such as that experienced during apnoea, is associated with Ca^2+^ overload, triggering ectopic activity. Prolonged exposure to atrial stretch is associated with chronic structural and inflammatory remodelling, providing a vulnerable substrate for re-entry [67]. Prevention of apnoeic atrial dilation in an obese rat model by balloon occlusion of the vena cava was shown to be associated with an 83% decrease in AF incidence during pacing [59]. In the same study, left atrial dilation during apnoea was significantly higher in obese animals, highlighting atrial stretch as a contributor to arrhythmogenesis in OSA, in which obesity is a common comorbidity. Furthermore, chronic left atrial enlargement and diastolic dysfunction are observed in numerous animal models of chronic OSA and are associated with a greater risk of AF recurrence in OSA patients [19,38,68,69]. Whilst the clinical evidence relating atrial stretch to arrhythmogenesis is strong, the mechanisms underpinning stretch-induced remodelling are particularly poorly characterised in the context of OSA and thus require further study. 

### 2.3. CIH

CIH is a central hallmark of OSA, with patients showing an average nadir oxygen desaturation of approximately 83% during sleep [70]. Whilst AHI remains the most common severity measure of OSA, other measures, such as total sleep time spent under 90% desaturation (T90), exist to quantify the degree of hypoxic burden. Interestingly, some studies report that T90 but not AHI is associated with AF incidence in OSA patients [36,71,72]. On the contrary, another large retrospective cohort study reported a strong, dose-dependent association with AHI [14]. 

A large body of evidence supports the view that CIH mediates significant atrial structural remodelling, adversely affecting AF inducibility. Atrial fibrosis is a common response to CIH, and this is consistent with reports of upregulated TGF-β, CTGF, and α-smooth muscle actin [39,45,46,47,48,56,73,74,75,76,77]. These pro-fibrotic events are closely linked to the inflammatory response, with markers such as TNF-α, IL-6, and IL-1β extensively reported in both animal models and OSA patients [64,70]. Attenuation of the inflammatory response in CIH-exposed rats was met with a concomitant decrease in pro-fibrotic markers and AF inducibility in response to liraglutide, a GLP-1 agonist with an emerging anti-inflammatory role [74]. The fibrotic substrate may be further aggravated by the remodelling of MMP—specifically, the upregulation of MMP-9 and the downregulation of MMP-2 [54,61,62]. Thus, CIH drives an imbalance between collagen secretion and degradation. CIH may also contribute to atrial remodelling by promoting inflammatory myocardial apoptosis [69,78]. These processes are summarised in Figure 1.

The electrical integrity of the atria is maintained, in part, by intercellular ion gap junctions, which play a key role in maintaining conduction velocity. CX-43 expression is reduced with CIH, with consequent increases in AF inducibility [38,68,79]. This may contribute to the reduced conduction velocity seen in other OSA models; however, conduction slowing secondary to CX dysregulation has not been explicitly demonstrated [47,73,74]. One study emphasised the importance of NADPH oxidase (NOX) 2 in CX remodelling, wherein NOX2-deficient mice abolished the CIH-induced downregulation of CX-40 and CX-43 [80]. This implies a reactive oxygen species (ROS)-dependent remodelling process, either by direct protein oxidation or secondary cell signalling processes. Indeed, markers of oxidative stress are elevated in numerous OSA models [56,57,78,79]. This is a prominent feature of CIH, as ROS are cyclically produced with intermittent re-oxygenation in a fashion similar to ischaemia–reperfusion injury [81]. Besides ROS signalling, hypoxia-inducible factor (HIF)-1α is known to play an important role in hypoxia sensing. Animal and cellular CIH models have demonstrated upregulations of HIF-1α with apoptosis, fibrosis, and inflammation [42,78]. Furthermore, serum HIF-1α is significantly upregulated in patients with OSA and may be useful as a biomarker of OSA severity [82,83]. This may be attributable, in part, to the ability of HIF-1α to upregulate inflammatory signalling via nuclear factor-κB, which is upregulated in a dose-dependent response with disease severity in OSA patients [84,85]. Overall, CIH contributes to structural remodelling in OSA through both hypoxic and inflammatory signalling pathways. These studies warrant the investigation of targeted antioxidant and anti-inflammatory treatment for the prevention of AF in OSA.

In addition to mediating local structural remodelling, CIH also acts to modify autonomic outflow through peripheral chemosensing, resulting in an increase in sympathetic nerve activity, as shown in Figure 1. It has become apparent that persistent pathological over-activation of the CB is responsible for the chronic rise in vascular sympathetic activity in both human patients with OSA and in animals exposed to CIH [56,86,87]. Data examining the role of a hyperactive CB in promoting rises in cardiac/atrial sympathetic activity after CIH are currently scarce. That said, it has been shown that CB ablation attenuates the rise in the HRV LF/HF ratio in rats exposed to CIH and significantly decreases spontaneous arrhythmia incidence [56]. The latter effect can be mimicked by treatment with propranolol, supporting the idea that CB-mediated arrhythmias are dependent on a heightened reflex sympathetic outflow to the heart. However, the specific cellular mechanisms of CB-mediated atrial arrhythmias remain elusive (especially in patients), and, clearly, the exploration of this area warrants future consideration. As well as augmenting activity, we have recently shown that CIH causes sympathetic hyperinnervation of the vasculature, an effect that is dependent on β-adrenoceptor stimulation and CB hyperactivity [88]. Whether or not left atrial sympathetic hyperinnervation and/or other pathological structural alterations observed after CIH are dependent on CB hyperactivity remains to be determined. A summary of the mechanisms associated with AF in OSA is presented in Figure 1.

## 3. Conclusions

Arrhythmogenesis in OSA is characterised by a complex interplay between numerous pathophysiological mechanisms. Atrial remodelling secondary to local hypoxia promotes oxidative and inflammatory stress, driving a vulnerable AF substrate through fibrosis, connexin remodelling, conduction slowing, cardiomyocyte apoptosis, and necrosis. Autonomic dysfunction is proposed to be driven by altered chemosensory and baroreceptor input and is exaggerated by the sensitisation of cardiomyocytes to sympathetic and parasympathetic activation. Acute apnoeic parasympathetic activation is likely to precipitate AF by contributing to AERP shortening. This is further enhanced by atrial stretch, driving AF induction through the dysregulation of Ca^2+^ handling and promoting further atrial structural remodelling. Future work should aim to characterise the cellular mechanisms underpinning local atrial remodelling and autonomic dysfunction as well as examine the sensitivity to currently available AADs. Existing evidence supports recommendations to screen for OSA in AF and ensure timely intervention as a means of preventing arrhythmogenesis. However, there remains a need for large, randomised control trials to investigate the effectiveness of CPAP for this use.

## Figures and Tables

**Figure 1 cells-12-01661-f001:**
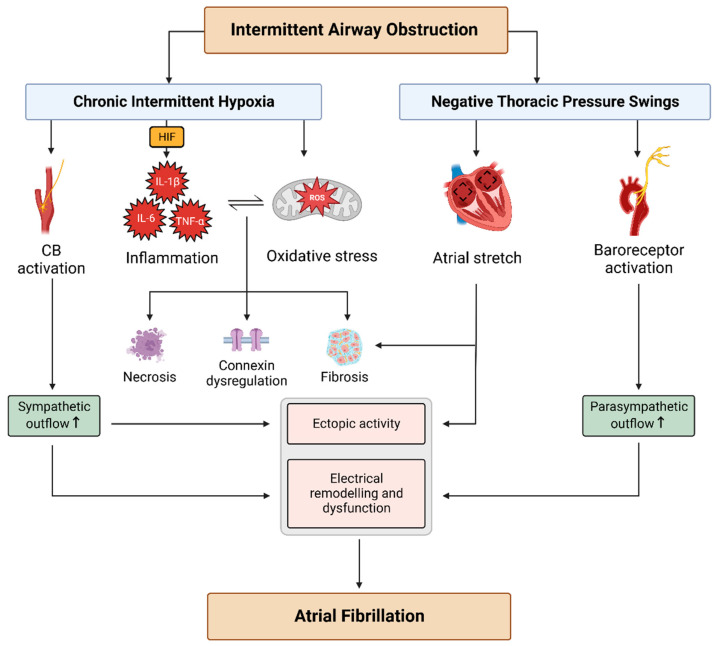
An overview of arrhythmogenic mechanisms in obstructive sleep apnoea. Chronic intermittent hypoxia secondary to intermittent airway obstruction promotes atrial structural remodelling through local and systemic inflammation as well as by mediating oxidative stress. Peripheral chemosensing activates sympathetic reflex activity to the heart, predisposing to ectopic activity and bringing about chronic alterations in cardiomyocyte ion channel expression. Negative thoracic pressure swings experienced during apnoea promote further structural remodelling and ectopic activity through atrial stretch. Baroreceptor activation causes acute AERP shortening, predisposing to re-entry. Abbreviations: ROS: reactive oxygen species; CB: carotid body; HIF: hypoxia-inducible factor; IL: interleukin; TNF-α: tumour necrosis factor-α.

## Data Availability

Not applicable.

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
