# Peer review of "Mechanisms of Atrial Fibrillation in Obstructive Sleep Apnoea"

_cells, 2023, doi:10.3390/cells12121661_

Round 1

Reviewer 1 Report

Better understanding of drug efficacy and genetic modification could improve rhythm control strategy.

OSA presented common risk factor, a dose-dependant response (AHI index and time under 90% SAO²), increasing risk of AF recurrence, CV event.

OSA induce cardiac remodelling, low voltage area and is related with poor autocomes in AF .

The need of integrative therapy of AF has driven to characterise mechanism of arrythmogeneicoty in OSA.

Three principal underlying mechanisms explain relationship between AF and OSA: Autonomic imbivalance, negative thoracic pressure and chronic intermittent hypoxia.

1.      Autonomic imbivalance: Chronic predominance in sympathetic tone secondary to hyperinnervation in OSA is met with a cellular adaptation in favour of counter balancing parasympathetic signalling.

Role of renin angiotensin system activation activated by sympathetic activity may also participate to atrial remodelling or CV outcome improving, which could decrease AF inducibility.

2.      Negative thoracic pressure during apnea has mechanistic effect and baroreceptor activation which favored left atrial stretch, which induced Ca2+ overload.

3.      Chronic intermittent hypoxia

Atrial fibrosis is a common response to hypoxia.l Clinical controversial studies are reported a dose dependant association between AF incidence and AHI.
Gap junction an CX 43 expression is decreased with CIH. ROS dependant remodelling process. HIF 1 alpha up regulation associated with apoptosis fibrosis inflammation.

CIH also plays a role in carotide body activation which seems to induce chronic rise in vascular sympathetic activity.

Author illustrated the manuscript with a comprehensive figure of the complexe interaction between AF and OSA.

Remark 1:Introduction:

Known physiological effect and definition of OSA has to be precise in introduction section.

Genetic modification are described in introduction but then are not specifically described in the text body. Either it should not be present in the introduction or specifically developed in the text body.

Same for antiarrhythmic drugs. It is discussed in the discussion but not developed in the text body. Relationship between OSA and antiarrhythmic drugs actions may be developed.

Introduction is a bit loung. It should be more concise and relevant.

Remark 2: Autonomic imbalance is supported by an autonomic hyperinnervation (histological). Is there any human histological studies which confirmed this finding?

Concerning the autonomic imbivalance in human, the typical heart rate variability in OSA patients  may be described for a better understanding. Especially the loss of the decrease HRV at night.

Furthermore, implication and chronology of the autonomic nervous system has been previously described(1) and may be described there to better understand underlying mechanism.

Over activation of the carotid body (last paragraph of the manuscript) , induced sympathetic nerve hyperactivity. What is the mechanism of carotid body activation? Is it a baroreceptor? Is it the same described line 160 and 254? Could it be precise by the authors? Is it also a part of the autonomic imbivalance?

Remark 3: (Figure 1):

Atrial fibrillation Tag should be add, and underlying mechanism of AF trigger and / or AF perpetuation may be add in the figure.

Same as in the body text: atrial remodelling is a vast definition. Kind of atrial remodelling may be precise.

Remark 4: Concerning the opening: “Existing evidence supports recommendations to screen for OSA in AF, and to ensure timely intervention as a means of preventing ar rhythmogenesis”. This is absolutely interesting but it may have been proposed independently in each part of the body text. How the diagnosis / treatment of the OSA may improve the arrhythmogenic effect?

Minor comments:

Line 44 : what does “aletered” means ? hyper or depolarization? Could the authors be more precise on this point.

Line 68, 72: what kinD of atrial remodelling? It has to be precise.

Line 71: low voltage: how was difined the low voltage. Where was it found: within the left atrium both atria, specifically within the posterieur wall or somewhere else? It has to be more precise.

Line 182: it should be said “illustrated” rather than “demonstrated” in figure 1.

Author Response

We thank the reviewer for their insightful comments and suggestions. We think that this has greatly helped to improve the manuscript and we value their input. 

Major Remarks

Known physiological effect and definition of OSA has to be precise in introduction section.

Thank you for pointing this out, a more concise definition of OSA has now been added along with some considerable information about its diagnosis and pathophysiology. The following text has been added along with two new references:

“Obstructive sleep apnoea (OSA) is a sleep-related breathing disorder characterised by intermittent upper airway collapse leading to recurrent apnoeas and hyponeas. Apneic events are classified as a greater than 90% reduction in baseline airflow for a period of 10 seconds or more, whilst hypopnoeas are defined as at least a 50% decrease in baseline airflow for a period of 10 seconds or more, accompanied by a 3-4% O2 desaturation or microarousal [3]. OSA severity is stratified based on the number of apnoeic/hypopneic events taking place per hour (apnoea hypopnoea index, AHI); with 5 to less than 15 classed as mild, 15 to 30 as moderate, and greater 30 as severe [3,4]. Key features of OSA include episodic exposure to hypoxia and hypercapnia, numerous microarousals, repetitive autonomic nervous system activation and large swings in intrathoracic pressure as the patient makes exaggerated respiratory efforts against an occluded upper airway [3].”

Genetic modification are described in introduction but then are not specifically described in the text body. Either it should not be present in the introduction or specifically developed in the text body. Same for antiarrhythmic drugs. It is discussed in the discussion but not developed in the text body. Relationship between OSA and antiarrhythmic drugs actions may be developed. Introduction is a bit loung. It should be more concise and relevant.”

Thank you for making this suggestion. We have now split the first section into two parts: 

1.1. OSA Predisposes to AF

1.2 Clinical Management of AF in Patients with OSA

Furthermore, the section on AADs has now been considerably expanded to consider the impact of OSA on AAD effectiveness and the potential interaction with genetic variants. It now reads as follows: 

“It has recently been shown that effective early rhythm control therapy (either ablation or AADs) decreases the risk of adverse cardiovascular outcomes in the long-term in patients with newly diagnosed AF [26]. Therefore, an emerging challenge is to achieve optimal and more personalized rhythm control therapy to ensure long term sinus rhythm. Many (10-20%) patients with AF are treated with AADs [2,27]. These medications are highly effective in some patients, but do not work in others and AF recurrence remains unacceptably high [2]. The variable efficacy likely reflects the numerous different underlying disease mechanisms that can lead to AF development and its progression [28]. An important research need is to better characterize the specific disease mechanisms underpinning AF development associated with genetic modifications and/or co-morbidities, including OSA.

Preclinical data has suggested that the effectiveness of flecainide and dronedarone can be predicted based on the expression level of key AF associated genes (e.g. PITX2) or the status of the atrial resting membrane potential, which is often more negative in AF patients [29,30]. At present, specific information regarding the efficacy of AADs in patients with AF and OSA is scarce. In a small cohort of 61 patients, it was observed that those with severe OSA were less responsive to AAD treatment compared to those without OSA or those with more mild/moderate OSA [31]. In a follow up pilot study, it was reported that the presence of AF-associated risk variants adjacent to the PITX2 gene reduced the effectiveness of AADs by the same amount as severe OSA [32]. Interestingly, the presence of these risk variants combined with severe OSA did not further deplete the sensitivity of the AADs, possibly indicating a shared pathway between the two. Large scale clinical trials are warranted to evaluate the effectiveness of individual AADs in OSA patients and to better understand the interaction between OSA and AF associated genetic variants. Furthermore, there is the potential to develop new AADs specifically for patients with OSA and AF, which are targeted to the underlying disease mechanism(s). This strategy should lead to better selection of AADs and more effective and personalized rhythm control therapy in patients with OSA and AF.

Autonomic imbalance is supported by an autonomic hyperinnervation (histological). Is there any human histological studies which confirmed this finding?

We agree that this is an interesting point. However, we are not aware of any human data from OSA patients with AF specifically looking at this. Although, some studies (Deneke et al. 2011 and Gould et al. 2006) have demonstrated atrial sympathetic hyperinnervation in a generalized population of AF patients (not OSA related). We have now included this to parallel the innervation density changes described in OSA:

“In AF patients, a similar atrial sympathetic hyperinnervation has been shown [37,38], with Deneke et al. also noting the downregulation in atrial cholinergic innervation. These studies demonstrate the role of autonomic imbalance in creating a vulnerable AF substrate.”

Concerning the autonomic imbalance in human, the typical heart rate variability in OSA patients may be described for a better understanding. Especially the loss of the decrease HRV at night.

We agree that HRV is a useful indicator for autonomic imbalance; we have now added a brief comment on the increase in LF/HF ratio seen in OSA patients during sleep (where we had originally only mentioned PRV changes).

Pulse/heart rate variability (PRV/HRV) are useful clinical measures of autonomic balance. Measures of HRV reveal a transition to sympathetic predominance in OSA patients during sleep, manifesting as increased low frequency / high frequency (LF/HF) ratio [31]. Other studies have found that increasing nocturnal PRV is an independent predictor of AF risk [32].”

Furthermore, implication and chronology of the autonomic nervous system has been previously described(1) and may be described there to better understand underlying mechanism.

We thank the reviewer for pointing this out. we agree that we have only really covered the long-term changes in cardiac autonomic balance and it is also important to consider the acute changes associated with the apnoeic episodes. We have therefore discussed the importance of this in the following section and added the accompanying references:

“Cardiac electrical function is subject to modification by the interplay between the sympathetic and parasympathetic arms of the autonomic nervous system. In OSA patients there are rapid transitions in cardiac autonomic balance which are dependent on the different phases of the acute apnoeic episodes. In the apnoeic period itself, there is a substantial bradycardia, which is indicative of parasympathetic dominance [33] Immediately after the apnoea, there is a powerful rebound tachycardia, which is likely due to parasympathetic withdrawal uncovering an increase in cardiac sympathetic activity. The heart rate then gradually returns back to resting levels [33]. Similarly, arterial blood pressure also rises immediately after the apnoea, suggestive of an acute rise in vascular sympathetic activity [34]. The precise mechanisms underpinning these changes in autonomic activity are poorly understood but are likely to have contributions from arterial baroreceptors, peripheral chemoreceptors, central respiratory centers, and pulmonary afferents. Furthermore, it is still not clear whether atrial arrhythmias are directly initiated by these acute changes in cardiac autonomic activity and if there is an increased risk either during or immediately after the apnoea.”

Atrial fibrillation Tag should be add, and underlying mechanism of AF trigger and / or AF perpetuation may be add in the figure.

An 'atrial fibrillation' tag has now been added to the figure. We would refrain from adding further mechanisms to the figure to avoid drawing attention away from main principles (hence the use of ' electrical remodelling' as a general term).

Concerning the opening: “Existing evidence supports recommendations to screen for OSA in AF, and to ensure timely intervention as a means of preventing arrhythmogenesis”. This is absolutely interesting but it may have been proposed independently in each part of the body text. How the diagnosis / treatment of the OSA may improve the arrhythmogenic effect?

We have rewritten the opening section and split this into two sections which now gives a better insight into the current management of AF in patients with OSA.

Minor

Line 44: what does “aletered” means ? hyper or depolarization? Could the authors be more precise on this point.

Thank you, yes this has now been clarified- Preclinical data has suggested that the effectiveness of flecainide and dronedarone can be predicted based on the expression level of key AF associated genes such as PITX2 or the status of the atrial resting membrane potential, which is often more negative in AF patients”

Line 68: what kind of atrial remodelling? It has to be precise.

Thank you, this has now been changed to “Patients with AF and OSA exhibit an increased risk of recurrences post ablation and increased prevalence of non-pulmonary vein triggers” 

Line 71/72: low voltage: how was difined the low voltage. Where was it found: within the left atrium both atria, specifically within the posterieur wall or somewhere else? It has to be more precise.

Thank you, this section has been expanded to include more specific information about the location, of type of electrical changes seen in patients with OSA and AF. It now reads as follows:

“The relationship between OSA and the AF substrate characterised by low left atrial voltage has been studied by Nalliah et al. using high density mapping [23]. They observed a dose-dependent association of OSA severity with the level of conduction heterogeneity, voltage heterogeneity and number of low voltage regions, which was consistent across all regions of the left atrium. In this study, low voltage was determined as a bipolar voltage of less than 0.5 mV. The AHI also associated with lower overall left atrial voltage, but not with slower conduction velocity. Patients with paroxysmal AF and OSA were suggested to have discrete low voltage zones combined with slow conduction velocity as the atrial substrate. In patients with persistent AF, there appeared to be a more diffuse pattern of low voltage across the entire left atrium”

Line 182: it should be said “illustrated” rather than “demonstrated” in figure 1.

Thank you this has been corrected

Reviewer 2 Report

Dear Authors,

There are some suggestions:

- Try to group references at the end of sentences rather than interspersing them within the text, as it hampers readability.

- Avoid overusing parentheses and excessively long sentence

- Define the acronyms the first time they appear, e.g., in line 235, “ROS”

- It is also necessary to define the acronyms in the caption of the figures.

- Define the acronyms consistently and only the first time they appear in the text - for example, T90 in lines 54 and 210.

- When using an acronym for a name, it is essential to use it consistently throughout the manuscript. For example, in line 35, 'antiarrhythmic drugs (AADs)' and line 38.

Please, review the English language carefully before the next delivery.

Author Response

We thank the reviewer for their important suggestions particularly regarding the clarity and readability. We think that the article is now much easier to follow.  

Reviewer 2:

Try to group references at the end of sentences rather than interspersing them within the text, as it hampers readability.

Thank you, we agree that this will make the article easier to read. The manuscript has been reformatted to ensure references follow at the end of sentences, e.g.:

“CPAP is an established means of reducing OSA severity, and has been shown to protect against AF incidence, recurrence, and progression, as well as alleviating antiarrhythmic drug (AAD) dependence in OSA patients [10,12,14,16-20].”

Avoid overusing parentheses and excessively long sentences

Thank you we have edited the manuscript and tried to remove parentheses where possible.

- Define the acronyms the first time they appear, e.g., in line 235, “ROS”

We have now defined the ROS acronym in line 285 of the new manuscript. We have also gone through the manuscript and defined abbreviations and acronyms the first time they appear. Thank you for pointing this out to us. 

- It is also necessary to define the acronyms in the caption of the figures.

Thank you. Abbreviations have now been added to the end of the figure legend:

“Abbreviations: ROS: reactive oxygen species; CB: carotid body; HIF: hypoxia-inducible factor; IL: interleukin; TNF-α: tumour necrosis factor-α.”

- Define the acronyms consistently and only the first time they appear in the text - for example, T90 in lines 54 and 210.

Thank you, we have gone through the manuscript and defined abbreviations and acronyms the first time they appear, including T90

When using an acronym for a name, it is essential to use it consistently throughout the manuscript. For example, in line 35, 'antiarrhythmic drugs (AADs)' and line 38.

The acronym for ‘antiarrhythmic drugs’ has now been defined in line 64 of the new manuscript, and used consistently throughout the text.

Reviewer 3 Report

The authors described that meta-analysis regarding CPAP therapy failed to show the benefit of CPAP to improve the outcome after catheter ablation for AF in patients with OSA. Are there any previous studies which demonstrated the efficacy of CPAP for primary prevention of AF in patients with OSA? Furthermore, to determine whether weight loss and/or exercise training decrease the incidence of coincident AF may add the supportive evidence of CPAP in patients with OSA. In this review manuscript, the authors provided important insights to readers in terms of the mechanism of obstructive sleep apnea, with broad coverage of the previous literature. Therefore, no further comments are present from the reviewer.

Author Response

Thank you for this important recommendation. We hope that this has now been addressed. 

Are there any previous studies which demonstrated the efficacy of CPAP for primary prevention of AF in patients with OSA? Furthermore, to determine whether weight loss and/or exercise training decrease the incidence of coincident AF may add the supportive evidence of CPAP in patients with OSA.

We were able to identify one study demonstrating that CPAP reduces arrhythmia in OSA patients without diagnosed AF (Abe et al. 2010 Heart and Vessels). This has been incorporated to the introduction section as evidence that CPAP may be useful for prevention of incident AF:

“CPAP is an established means of reducing OSA severity, and has been shown to protect against AF incidence, recurrence, and progression, as well as alleviating antiarrhythmic drug (AAD) dependence in OSA patients [8,10,12,14-18].”